# Evaluation of Generative Models: An Empirical Study

## Abstract

Implicit generative models, which do not return likelihood values, such as generative adversarial networks and diffusion models, have become prevalent in recent years. While it's true that these models have shown remarkable results, evaluating their performance is challenging. This issue is of vital importance to push research forward and identify meaningful gains from random noise. Currently, heuristic metrics such as the Inception score (IS) and Fréchet Inception Distance (FID) are the most common evaluation metrics, but what they measure is not entirely clear. Additionally, there are questions regarding how meaningful their score actually is. In this work, we propose a novel evaluation protocol for likelihood-based generative models, based on generating a high-quality synthetic dataset on which we can estimate classical metrics for comparison. Our study shows that while FID and IS do correlate to several f-divergences, their ranking of close models can vary considerably making them problematic when used for fine-grained comparison. We further used this experimental setting to study which evaluation metric best correlates with our probabilistic metrics. Lastly, we also address some of the issues with FID score by investigating the features used for this metric.

## 1 Introduction

Implicit generative models such as Generative Adversarial Networks (GANs) Goodfellow et al. (2014) have made significant progress in recent years, and are capable of generating high-quality images Karras et al. (2020b); Ramesh et al. (2022) and audio Kong et al. (2020). Despite these successes, evaluation is still a major challenge for implicit models that do not predict likelihood values. While significant improvement can easily be observed visually, at least for images, an empirical measure is required as an objective criterion and for comparison between relatively similar models. Moreover, devising objective criteria is vital for development, where one must choose between several design choices, hyperparameters, etc. The most common practice is to use metrics such as Inception score (IS) Salimans et al. (2016) and Fréchet Inception Distance (FID) Heusel et al. (2018) that are based on features and scores computed using a network pre-trained on the ImageNet Deng et al. (2009) dataset. While these proved to be valuable tools, they have some key limitations: (i) It is unclear how they relate to any classical metrics on probabilistic spaces. (ii) These metrics are based on features and classification scores trained on a certain dataset and image size, and it is not clear how well they transfer to other image types, e.g. human faces, and image sizes, (iii) The scores can heavily depend on particular implementation details Barratt & Sharma (2018b); Parmar et al. (2021).

Another evaluation tool is querying humans. One can ask multiple human annotators to classify an image as real or fake or to state which of two images they prefer. While this metric directly measures what we commonly care about in most applications, it requires a costly and time-consuming evaluation phase. Another issue with this metric is that it does not measure diversity, as returning a single good output can get a good score.

In this article we offer a new evaluation protocol for likelihood-based generative models such as Auto-Regressive (AR) and Variational Auto-Encoders (VAE) Kingma & Welling (2014). We created a high-quality synthetic dataset, using the powerful Image-GPT model Chen et al. (2020). This is a complex synthetic data distribution that we can sample from and compute exact likelihood values. As this data distribution is trained on natural images from the ImageNet dataset using a strong model, we expect the findings on it to

be relevant to models trained on real images. The dataset provides a solid and useful test-bed for developing and experimenting with generative models. We will make our dataset public for further research[1].

Using this test-bed we train various likelihood models and evaluate their KL-divergence and reverse KL-divergence. While our interest is implicit models, we experiment with likelihood models as they have alternative well-understood metrics for comparison. This allows us to compare the well-understood divergences to empirical metrics such as FID and evaluate their capabilities. We expect our results to transfer to implicit models as metrics such as FID and IS are not tailored to a specific kind of model. We observe that while the empirical metrics correlate nicely to these divergences, they are much more volatile and thus might not be well-suited for fine-grained comparison.

Finally, we investigated the use of the Inception network Szegedy et al. (2015) for feature extraction on FID, specifically for image datasets that are different from the ImageNet on which it was trained. This is important as FID is commonly used to compare models on datasets such as CelebA (human faces) Liu et al. (2015) and LSUN Yu et al. (2015) bedrooms that are quite distinct from ImageNet. Specifically, we investigate the Gaussianity assumption that lies in the base of the FID metric, compared to features returned by CLIP Radford et al. (2021) which was trained on a wider variety of images. We show both quantitatively and qualitatively that the CLIP features are better suited than the Inception features on non-ImageNet datasets.

## 2 Background

Given the popularity of GANs and other implicit generative models, many heuristic evaluation metrics have been proposed in recent years. We give a quick overview of the most common metrics and probabilistic KL-divergences.

### 2.1 $KL$-Divergence

One common measure of the difference between probability distributions is the Kullback–Leibler (KL) divergence $KL(p||q) = \mathbb{E}_{x \sim p}\left[\log\left(\frac{p(x)}{q(x)}\right)\right]$, noting that it is not symmetric. We refer to $KL(p_{data}||p_{model})$ as the KL divergence and $KL(p_{model}||p_{data})$ as the Reverse KL (RKL) divergence, where $p_{data}$ denote the real data distribution, and $p_{model}$ denote the approximated distribution, learned by the generative model. Minimizing the log-likelihood is the same as minimizing the KL divergence between $p_{data}$ and $p_{model}$ up to a constant, hence it can be done even when $p_{data}$ is unknown. It is important to note that the KL divergence is biased towards "inclusive" models where the model "covers" all high-likelihood areas of the data distribution and punishes harder when $p_{data}(x) \gg p_{model}(x)$ (figure 1, left). The RKL has a bias toward "exclusive" models, where the model does not cover low likelihood areas of the data distribution and punishes harder when $p_{data}(x) \ll p_{model}(x)$ (figure 1, right). While an exclusive bias might be more appropriate in some applications, such as out-of-distribution detection, we cannot optimize it directly without access to $p_{data}$. As these divergences measure complementary aspects, we believe that examining both of them simultaneously gives us a well-rounded view of the generative model behavior. A limitation of KL divergence is that it does not consider the metric properties of the sample space, as opposed to Wasserstein distance, therefore it is less suitable for GAN training since it uses samples directly in the training process Arjovsky et al. (2017).

---

[1]Link will be added to the final version for anonymity

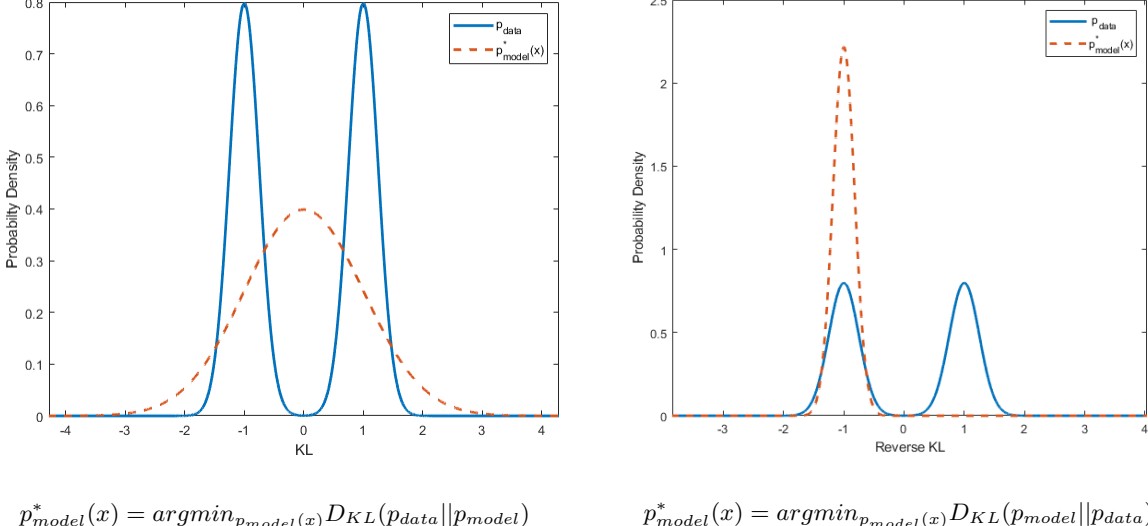

$$p_{model}^*(x) = argmin_{p_{model}(x)}D_{KL}(p_{data}||p_{model}) \qquad p_{model}^*(x) = argmin_{p_{model}(x)}D_{KL}(p_{model}||p_{data})$$

Figure 1: Optimizing $p_{model}$ with KL criteria pushes the model to cover all aspects of $p_{data}$, hence it is more exclusive while optimizing it with reverse KL criteria encourages the model to cover the area with the largest probability, hence it is more inclusive.

## 2.2 Inception Score

Inception Score (IS) is a metric for evaluating the quality of image generative models based on InceptionV3 Network pre-trained on ImageNet. It calculates:

$$IS = \exp\left(E_{x \sim p_{model}}[KL(p_\theta(y|x)||p_\theta(y)])\right)$$

where $x \sim p_{model}$ is a generated image, $p_\theta(y|x)$ is the conditional class distribution computed via the inception network, and $p_\theta(y) = \int_x p_\theta(y|x)p_{model}(x)dx$ is the marginal class distribution.

The two desired qualities that this metric aims to capture are: (i) The generative model should output a diverse set of images from all the different classes in ImageNet, i.e $p_\theta(y)$ should be uniform (ii) The images generated should contain clear objects so the predicted probabilities $p_\theta(y|x)$ should be close to a one-hot vector and have low entropy. When both of this qualities are satisfied then the KL distance between $p_\theta(y)$ and $p_\theta(y|x)$ is maximized. Therefore the higher the IS is, the better.

## 2.3 Fréchet Inception Distance

The FID metric is based on the assumption that the features computed by a pre-trained Inception network, for both real and generated images, have a Gaussian distribution. We can then use known metrics for Gaussians as our distance metric. Specifically, FID uses the Fréchet distance between two multivariate Gaussians which has a closed-form formula. For both real and generated images we fit Gaussian distributions to the features extracted by the inception network at the pool3 layer and compute

$$FID = ||\mu_r - \mu_g||^2 + Tr(\Sigma_r + \Sigma_g - 2(\Sigma_r\Sigma_g)^{1/2})$$

where $\mathcal{N}(\mu_r, \Sigma_r)$ and $\mathcal{N}(\mu_g, \Sigma_g)$ are the Gaussian fitted to the real and generated data respectively. The quality of this metric depends on the features returned by the inception net, how informative are they about the image quality, and how reasonable is the Gaussian assumption about them.

## 2.4 Kernel Inception Distance

The Kernel Inception Distance (KID) Bińkowski et al. (2018) aims to improve on FID by relaxing the Gaussian assumption. KID measures the squared Maximum Mean Discrepancy (MMD) between the Inception

representations of the real and generated samples using a polynomial kernel. This is a non-parametric test so it does not have the strict Gaussian assumption, only assuming that the kernel is a good similarity measure. It also requires fewer samples as we do not need to fit the quadratic covariance matrix. The motivation for this is the bias of the FID and IS.

## 2.5 FID$_\infty$ & IS$_\infty$

In Chong & Forsyth (2020) the authors show that the FID and IS metrics are biased when they are estimated from samples and that this bias depends on the model. As the bias is model-dependent, it can skew the comparison between different models. The authors then propose unbiased version of FID and IS named FID$_\infty$ / IS$_\infty$.

## 2.6 Clean FID

As the input to the Inception network is fix-sized, generated images of different sizes need to be resized to fit the network's desired input dimension. The work in Parmar et al. (2022) investigates the effect of this resizing on the FID score, as the resizing can cause aliasing artifacts. The lack of consistency in the processing method can lead to different FID scores, regardless of the generative model capabilities. They introduce a unified process that has the best performance in terms of image processing quality and provide a public framework for evaluation.

## 2.7 Ranking Correlation Methods

To compare the different scoring methods, we will evaluate how they differ in ranking different models. This allows us to focus on their main purpose of ranking different models. To do that we will use ranking correlation metrics.

### 2.7.1 Spearman Correlation

The Spearman correlation coefficient (Spearman, 1904) is defined as the Pearson correlation coefficient between the rank variables. For $n$ elements being ranked, the raw scores $X_i, Y_i$ are converted to ranks $R(X_i), R(Y_i)$. The Spearman correlation coefficient $r_s$ is defined as:

$$r_s = \rho_{\mathrm{R}(X),\mathrm{R}(Y)} = \frac{\mathrm{cov}(\mathrm{R}(X),\mathrm{R}(Y))}{\sigma_{\mathrm{R}(X)}\sigma_{\mathrm{R}(Y)}}$$

$\rho$ denotes the usual Pearson correlation coefficient, but applied to the rank variables, $\mathrm{cov}(\mathrm{R}(X),\mathrm{R}(Y))$ is the covariance of the rank variables, $\sigma_{\mathrm{R}(X)}$ and $\sigma_{\mathrm{R}(Y)}$ are the standard deviations of the rank variables.

### 2.7.2 Kendall's $\tau$

Kendall's KENDALL (1938) correlation coefficient assesses the strength of association between pairs of observations based on the patterns of concordance and discordance between them. A consistent order (concordance) is when $x_2 - x_1$ and $y_2 - y_1$ have the same sign.
Inconsistently ordered (discordant) occurs when a pair of observations is concordant if $x_2 - x_1$ and $y_2 - y_1$ have opposite signs. Kendall's $\tau$ is defined as $\tau = \frac{\mathbf{C}-\mathbf{DC}}{\binom{n}{2}}$,

where $\mathbf{C}$ is the number of concordance pairs in the list and $\mathbf{DC}$ is the number of discordant.

## 2.8 Related Works

In addition to previously mentioned works that defined empirical metrics, other works looked into the evaluation of generative models. Bond-Taylor et al. (2021) performed a comparative review of deep generative models. Borji (2021) presents a comprehensive survey of generative model estimation methods. In Theis et al. (2016) investigae likehood based models and show on a toy examples how independent different evaluation

methods are. We support this thesis and perform a thorough empirical study on actual datasets and compare the latest generative models evaluation methods. Barratt & Sharma (2018a) first pointed out issues in IS. Lee & Lee (2021) inspect the distribution of the Inception latent feature and suggest a more accurate model for evaluation purposes. Xu et al. (2018) perform an empirical study on an older class of evaluation metrics of GANs and mention that KID outperforms FID and IS. Fedus et al. (2018) shows IS high sensitivity to the dataset trained by the backbone network (in this example, ImageNet and CIFAR-10). Lucic et al. (2018) shows FID sensitivity to layers and features of the backbone network and for mode dropping.

Another line of works by Shmelkov et al. (2018); Lesort et al. (2019); Santurkar et al. (2018); Ravuri & Vinyals (2019) utilize the classification score of generated data to evaluate models performances. Despite its usefulness, a classification score is not foolproof. During adversarial attacks, for example, the image may appear perfect, but its classification score will be poor.

The latest works propose precision and recall as a way to disentangle the quality of generated samples from the coverage of the target distribution Sajjadi et al. (2018); Kynkäänniemi et al. (2019).

## 3  Method

As the first step of our method, we train an auto-regressive model to approximate the information distribution. Using the model, whose distribution we know, we create a high-quality synthetic dataset and then examine the performance of other likelihood-based models against the synthetic data. The following are the steps involved in the method:

---

**Algorithm 1** Creating Synthetic Dataset With Known Likelihood

---

1: Train likelihood-based generative model[1] on dataset $X$
2: Generate $\hat{X}$, N samples from $p_{data}(x)$ with known likelihood
3: Split $\hat{X}$ to train set and test set
4: Train likelihood-based generative model[2] with the train set
5: Evaluate $p_{model}(\hat{X})$ on test set from model[2]
6: Measure $KL(p_{data}(\hat{X})||p_{model}(\hat{X}))$ and $KL(p_{model}(\hat{X})||p_{data}(\hat{X}))$ on test set

---

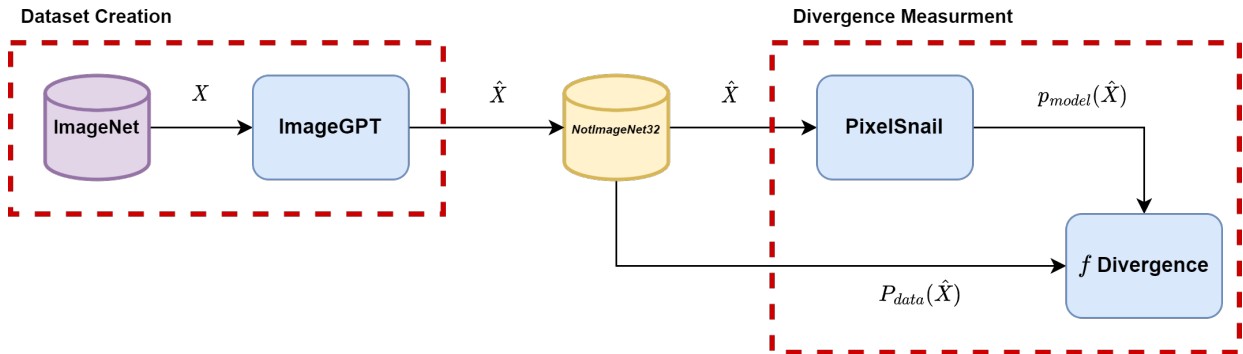

Figure 2: Illustration: $X$ are ImageNet images, $\hat{X}$ are synthetic images that sampled from image-GPT, $p_{data}(\hat{X})$ is ground truth likelihood from image-GPT for synthetic images and $p_{model}(\hat{X})$ is likelihood estimation of $p_{data}(\hat{X})$, calculated by the evaluated model, in this case, PixelSnail.

In this article, we created an auxiliary, realistic, dataset by sampling images from the Image-GPT model that has been trained on ImageNet32, the ImageNet dataset that was resized to $32 \times 32$. ImageGPT was chosen as a reference for being a powerful AR model with 1M epochs training checkpoints available [2]. We split the data set into a training set (70K images) and a test set (30K images), similar in size to CIFAR10, a common

---

[2]https://github.com/openai/image-gpt

benchmark. ImageGPT's ability to generate quality and realistic samples is demonstrated qualitatively in Fig. 3 and quantitatively by the high results in linear probability scores. As this is a synthetic version of ImageNet32 we name our dataset ***NotImageNet32***.

We note that Image-GPT clusters the RGB values of each pixel into 512 clusters and predicts these cluster indexes. This means that instead of each pixel corresponding to an element of $\{0, ..., 255\}^3$ it belongs to $\{0, ..., 511\}$. We can map these cluster values back to RGB, as was done in Image-GPT, for visualization.

This scheme is not restricted to *NotImageNet32*, Which is brought as an example for a single use case. In general, we advocate for using high-quality synthetic datasets to bridge the gap between real data on which performance is hard to evaluate and toy problems that do not necessarily represent real challenges. This can be utilized for ranking State of The Art (SOTA) generative models and finding hyperparameters of the data generation process such that they produce the least amount of inconsistencies across measurements.

To evaluate and understand current heuristic generative model metrics we train a set of models on *NotImageNet32*. One set of models is based on the PixelSnail model Chen et al. (2017). We use PixelSnail as it is a strong Autoregressive model, but not as powerful as the pixel-GPT that generated the data. From this we expect it to be able to fit the data well, but not perfectly. For diversity, we also measure a VAE model, based on VD-VAE Child (2021) (we used IWAE Burda et al. (2016) to reduce the gap between the ELBO and the actual likelihood). We note that all models were adjusted to our dataset and output the clustered index instead of RGB values. Supplementary details on the models architecture in this experiment can be found in the appendix section.

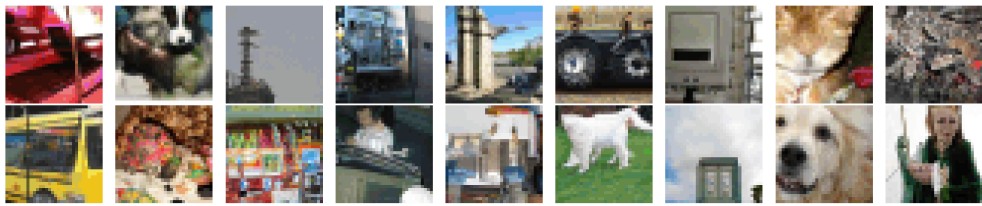

Figure 3: Examples of photos that are generated by image-GPT. Each photo's explicit likelihood can be measured.

To produce a diverse set of models with varying degrees of quality, each set was trained several times with different model sizes. We save a model for comparison after every five epochs of training. As a result, the models we compare are a mix of strong and weak models. After the training procedure, we can compute for each image in the test set its likelihood score (or the IWAE bound) for each model.

We then measure the difference between $p_{data}(x)$ and $p_{model}(x)$ by using Monte-Carlo approximation of two divergence function: Kullback–Leibler (KL) $KL(p_{data}||p_{model})$ and Reverse KL (RKL) $KL(p_{model}||p_{data})$. As these divergences measure complementary aspects, one inclusive and one exclusive, we believe that this, although unable to capture all the complexities of a generative model, gives us a well-rounded view of the generative model behavior. KL-Divergence has been thoroughly investigated in the fields of probability and information theory, and their properties along with what they measure are well known. Thus, comparing them to heuristic methods such as FID will shed light on these empirical methods.

A limitation of this test bed is that it can be applied only to likelihood-based models, so implicit models like GAN are not able to take advantage of it.

## 4   Comparison Between Evaluation Metrics

### 4.1   Volatility

We first train four PixelSnail variants on our NotImageNet32 dataset and plot the KL, RKL, FID, and IS (we plot the negative IS so lower is better for all metrics) along with the training for test set in Fig. 4 and 5. It can easily be seen that after 15-20 epochs both KL and RKL change slowly, but the FID and IS are

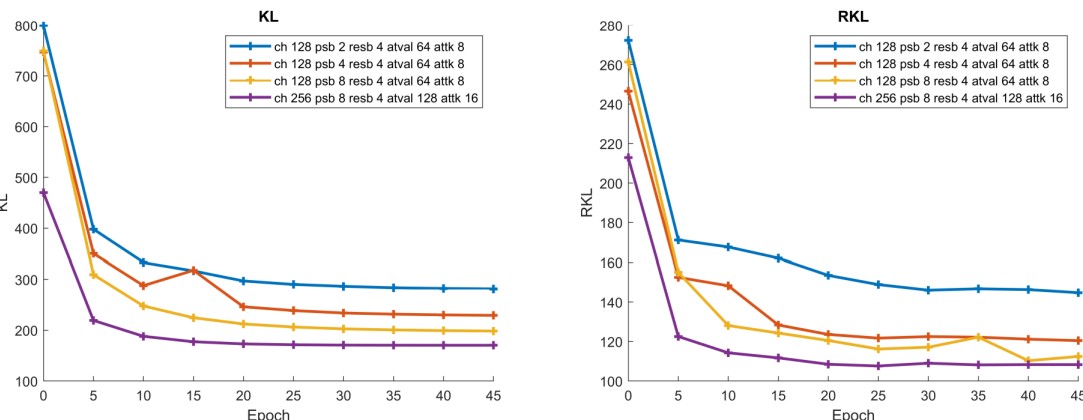

Figure 4: Test KL and RKL of PixelSnail models along training.

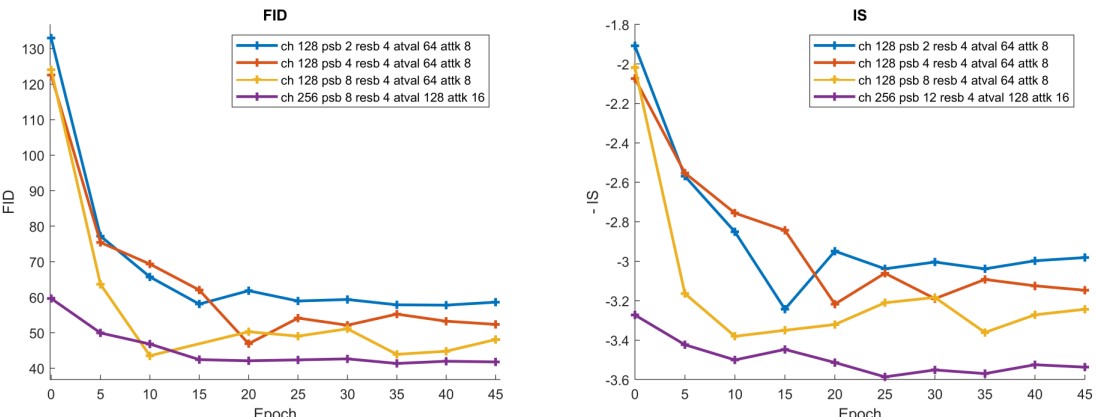

Figure 5: Test FID and negative IS of PixelSnail models along training. We plot the negative Inception Score so lower is better for all metrics. Details on the hyperparameters summerized in the legend are in the appendix.

much more volatile. Each dot in the graph represents a score that has been measured on a different epoch on a different model. To assess the variance of the results we used the Jack-Knife resampling method Tukey (1958). The error bar was small ($10^{-3}$ scale in most cases), hence it was unnoticeable. One can see from this figure that as we increase the model capacity, the KL score improves. Model-generated samples are included in appendix E. models Interestingly, the KL and RKL have a high agreement even if they penalize very different mistakes in the model. In stark contrast, we see that the FID, and especially IS, are much more volatile and can give very different scores to models that have very similar KL and RKL scores.

To get another perspective, we plot in Fig. 6 the FID and negative IS vs. KL and RKL. We observe a high correlation between FID/IS and KL and a weaker correlation between these metrics and the RKL. IS and FID are also seem ill-suited for fine-grained comparisons between models. For high-quality models, e.g., light-blue dots in Fig. 6, one can get a significant change in FID/IS without a significant change to KL/RKL. This can be very problematic, as when comparing similar models, e.g. testing various design choices, these metrics can imply significant improvement even when it is not seen in our probabilistic metrics. We add zoomed-in versions of this plot to appendix A for greater clarity.

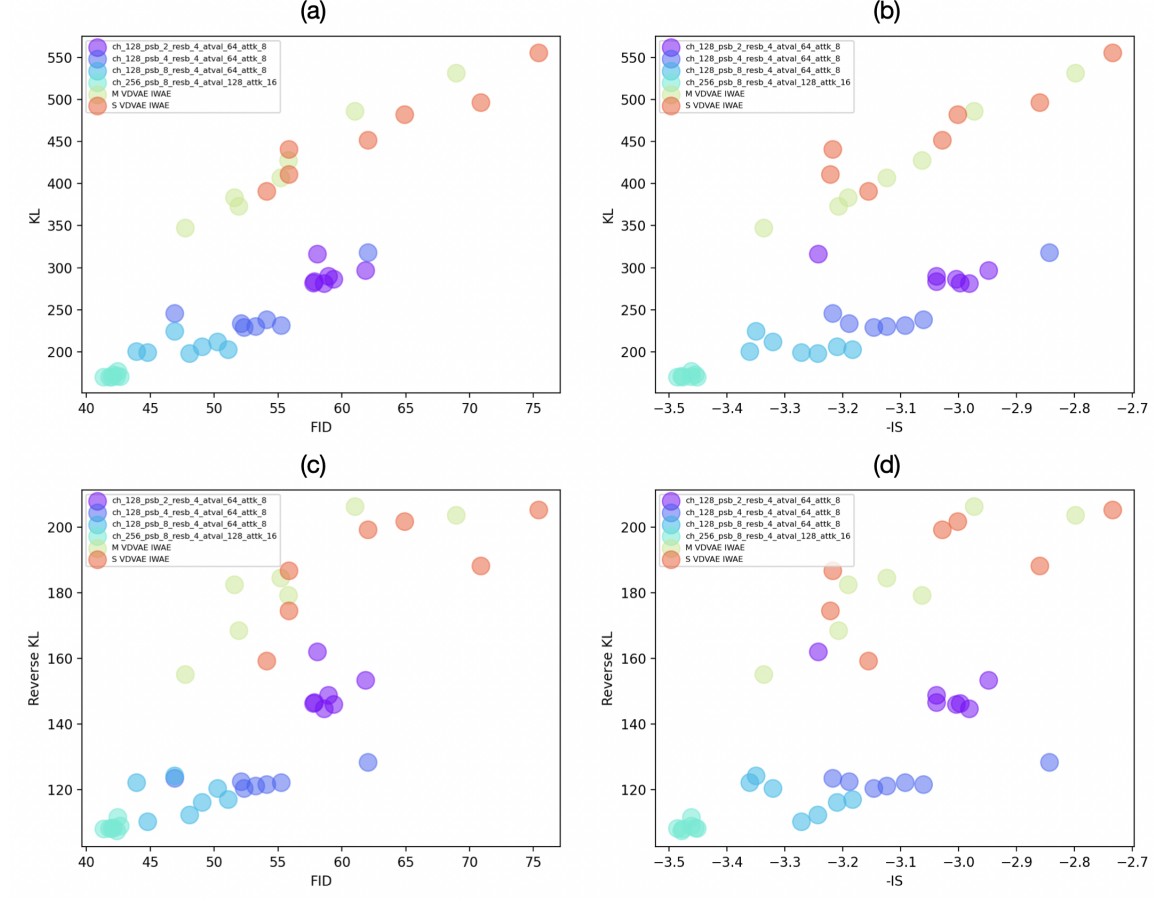

Figure 6: Evaluation metrics along the training of four pixelsnail and two VD-VAE models of varying sizes. We plot the negative Inception Score so lower is better for all metrics.

## 4.2 Ranking Correlation

To better quantitatively assess our previous observations, we compare how the metrics differ in their ranking of the various trained models. This is of great importance, as comparing different models is the primary goal of these metrics. To compare the ranking we compute Kendall's $\tau$ ranking correlation (Tab. 1). We perform the correlation analysis for models that were trained for 15 - 45 epochs and ignore the first iterations of the training procedure. This is done to focus more on the fine-grained comparisons.

Table 1: Kendall's $\tau$ correlation between different metrics. A correlation score indicates the degree of agreement between two scoring methods.

|  | KL | RKL | FID | IS | $IS_\infty$ | KID | $FID_\infty$ | Clean FID |
|---|---|---|---|---|---|---|---|---|
| KL | 1 | **0.8895** | 0.7027 | 0.5889 | 0.4681 | 0.7770 | 0.8095 | 0.7909 |
| RKL | **0.8895** | 1 | 0.6337 | 0.5244 | 0.4314 | 0.7105 | 0.7267 | 0.7198 |
| FID | 0.7027 | 0.6337 | 1 | 0.7979 | 0.7189 | 0.8513 | 0.8002 | 0.8699 |
| IS | 0.5889 | 0.5244 | 0.7979 | 1 | 0.8281 | 0.7329 | 0.6818 | 0.7236 |
| $IS_\infty$ | 0.4681 | 0.4314 | 0.7189 | 0.8281 | 1 | 0.6167 | 0.5749 | 0.6074 |
| KID | 0.7770 | 0.7105 | 0.8513 | 0.7329 | 0.6167 | 1 | 0.8606 | 0.9675 |
| $FID_\infty$ | 0.8095 | 0.7267 | 0.8002 | 0.6818 | 0.5749 | 0.8606 | 1 | 0.8746 |
| Clean FID | 0.7909 | 0.7198 | 0.8699 | 0.7236 | 0.6074 | 0.9675 | 0.8746 | 1 |

The highest score in both ranking correlation methods is between KL and Reverse KL with 0.889 Kendall's $\tau$. This may be surprising since these two methods measure different characteristics of the data. Confirming our previous observation, the FID and IS ranking scores are low, with FID outperforming IS. However, the extensions of FID do achieve better scores.

Another observation is the relatively low correlation between many of the different rankings. All of the Inception ranking correlation, except one (KID and Clean FID), indicates that one can get significantly different rankings by using a different metric.

Among the Inception-based metrics, $FID_\infty$ has the highest correlation with KL and RKL which indicates that it is a more reliable metric than the other. $IS/IS_\infty$ has the lowest ranking correlation between all other models.

## 5 Is Inception All We Need?

In the previous section, we evaluated the performance of FID and IS and found issues with them. We will now investigate one potential issue with these metrics, the backbone Inception network. Most common metrics are based on features computed by a pre-trained Inception network trained on the ImageNet classification task. The underlying assumption behind FID and its extensions is that these features are representative of the quality of the image and that they follow a Gaussian distribution.

As these metrics are used to evaluate generative models on various domains, e.g., faces, pets, bedrooms, etc., that are distinct from the ImageNet dataset on which the Inception network was trained, it raises the question: Are the features returned by the Inception network the right choice for comparing generative models in general?

In the next section, we evaluate the Inception features qualitatively and compare them to the features computed by the CLIP (Contrastive Language-Image Pre-Training) network. Additional quantitative experiments are in the supplementary material. CLIP is a neural network trained on the task of matching images to captions. It was trained on 400M images from a wide variety of domains, and was shown in multiple works to give strong representations that are useful for generating images Gal et al. (2022); Galatolo et al. (2021); Zhou et al. (2021). We hypothesize that since CLIP was trained on multiple domains and using full image captions, its features would be better suited for comparing generative models.

### 5.1 Qualitative Analysis

FID and its extensions are based on the assumption that the distribution over the latent representation is Gaussian. Here we evaluate how this Gaussian assumption holds. To do this we fit a Gaussian to the real data using each representation and look at the generated images that get the best/worst likelihood according to this Gaussian. In detail we:

1. Sample 10K images from a generative model.

2. Randomly select 20K images from the original data set used to train the generative model and compute their feature vectors with the Inception network and the CLIP network.

3. For each of these representations, fit a Gaussian model.

4. Calculate the probability of each of the synthetic samples belonging to the corresponded Gaussian model and rank them by their score.

In Fig. 7 we show the images that got the lowest probability rank on the AFHQ dataset Choi et al. (2019) with the wild class. We used a pre-trained StyleGAN2-ADA Karras et al. (2020a) as our generative model.

Since those images got the lowest probability rank among 10K images, our Gaussian model classifies them as outliers. If matching the Gaussian on these features is a good metric, then these low-probability images should correspond to low-quality generated images. As one can see, many of the CLIP low-probability images

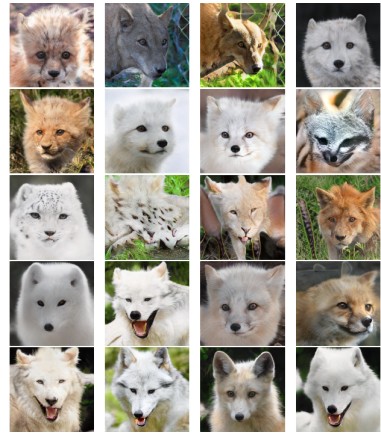
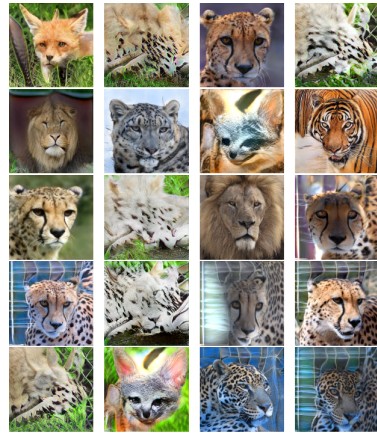

Inception - low probability (Wild).          CLIP - low probability (Wild).

Figure 7: Samples that were classified as outliers while modeling the latent variables as Gaussian model. Inception samples do not look like outliers, unlike CLIP samples, which are distorted.

look indeed like anomalies while most of the Inception anomalies look like valid images from Wild dataset. While this is only a small number of images, we see similar behavior across various non-ImageNet datasets and generative models. These results indicate that CLIP features are better suited for comparison between generative models. More results on different datasets are available in appendix D.

## 5.2 Normality Test

We augment our previous qualitative assessment by testing how well our samples follow a Gaussian distribution on our features. To utilize the readily available normality tests on 1D data, we linearly project our data randomly to one dimension and use these tests on multiple projections. This is valid, as a linear mapping of a multivariate normal also follows a normal distribution. We point out that we chose to analyze the normality test in a single dimension, rather than in a multi-dimension, for numerical stability considerations. Specifically, we:

1. Propagate dataset with $N$ pictures via Inception and CLIP and save the latent vectors of the images. $A \in \mathbb{R}^{N \times d}$ is the result matrix, where $d$ is the latent representation dimension (2048 for Inception and 512 for CLIP).

2. Generate $\mathbf{x} \in \mathbb{R}^d$ unit vector in a uniformly random direction.

3. Calculate $\mathbf{z} = A\mathbf{x} \in \mathbb{R}^N$, The projection of $A$ on random direction $\mathbf{x}$.

4. Run the D'Agostino's K-squared normality test D'Agostino & Pearson (1973) and calculate $p$ value under the null hypothesis that the data were drawn from a Gaussian distribution.

5. Repeat the process for $T = 1000$ times for different randomized unit vectors and calculate the mean of $p$ value.

The results, reported in Table 2, indicate that the Inception features are non-Gaussian and that across the board CLIP achieves better scores. Surprisingly, Inception features achieved the best score, by far, on our synthetic dataset and not on the original ImageNet dataset on which they were trained. We hypothesize that this is because our images were also generated by a deep neural network, albeit with a much different structure than the Inception network, and thus share certain characteristics as a result.

Table 2: Mean $p$ value results of normality test. The lines represent the normality scores for CLIP and Inception latent variables across datasets.

|  | CLIP | Inception |
|---|---|---|
| **AFHQ -Wild** | 0.0162 | 4.07E-192 |
| **AFHQ -Dogs** | 0.1893 | 1.40E-31 |
| **CelebA** | 0.0674 | 2.10E-20 |
| **NotImageNet32** | 0.1328 | 0.0049 |
| **ImageNet** | 0.093 | 6.34E-59 |

## 6 Conclusions

To summarise, we generated a high-quality synthetic dataset and compared the standard empirical metrics such as FID and IS to probabilistic f-divergences such as KL and RKL. We first observe that the empirical metrics show good correlation, so they do capture important trends. However, they are much more volatile and not all significant gains in one of the metrics correspond to observable gains in one of the KL divergences. We also observed that IS and its IS∞ extensions performed significantly worse compared to all other metrics. Finally, we investigated the standard use of the Inception features and show that, especially on benchmarks that are not ImageNet, they are outperformed by the more general-purpose CLIP features.

Given these observations we recommend:

- Drop the use of Inception Score, and used $\text{FID}_\infty$ instead of FID.

- Use multiple metrics (e.g. $\text{FID}_\infty$, KID and Clear FID) to try and control the volatility in scores.

- Replace the inception network with CLIP in FID. We made the code for FID based on CLIP available. Link will be added to final version for anonymity.

- Advocate NotImageNet32 as test-bed for generative models.

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

# A Volatility Analysis of High-Quality Models

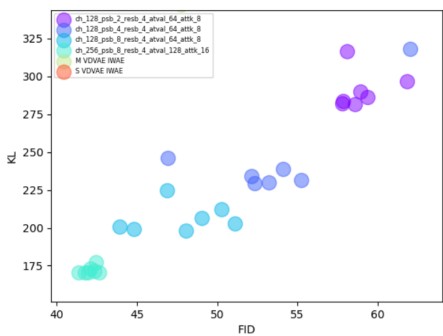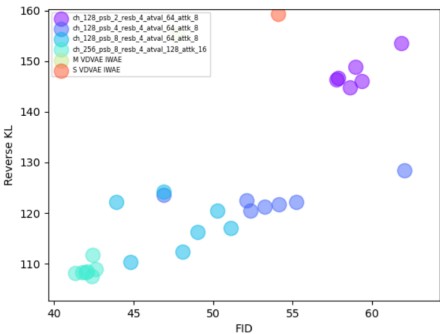

Figure 8: Evaluation metrics along the training of four PixelSnail and two VD-VAE models of varying sizes. Zoom in on high-quality models.

In Fig 8 one can see that FID score dramatically changes although there is not much change in the KL or in the RKL metrics. This may indicate on the volatility of this method.

# B Technical details on experiment's generative models architecture

As mentioned in section 4, we create different models by setting different hyper-parameters in order to compare performances between them. In order to enable accurate reproduction capability, we describe the set of parameters we used.

## B.1 PixelSnail

The PixelSNAIL architecture is primarily composed of two main components: residual block, which applies several 2D-convolutions to its input, each with residual connections. The other is the attention block, which performs a single key-value lookup. It projects the input to a lower dimensionality to produce the keys and values and then uses softmax-attention. The model is built from serval PixelSnail blocks concat to one another, each interleaves the residual blocks and attention blocks mentioned earlier. We used Adam optimizer with LR 0.0001 and MultiplicativeLR scheduler with lambada LR 0.999977. The loss function changed to the mean cross-entropy over 512 discrete clusters. All the other parameters that make up a model are described in table 3.

Table 3: PixelSnail hyper-parameters

| Size | Channels | PixelSnail blocks | Residual blocks | Attention values | Attention keys |
|------|----------|-------------------|-----------------|------------------|----------------|
| S    | 128      | 2                 | 4               | 64               | 8              |
| M    | 128      | 4                 | 4               | 64               | 8              |
| L    | 128      | 8                 | 4               | 64               | 8              |
| XL   | 256      | 8                 | 4               | 128              | 16             |

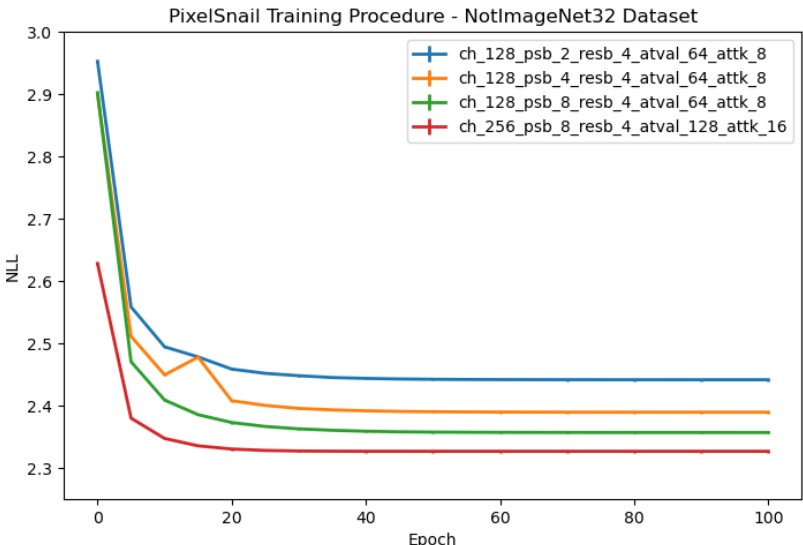

Figure 9: NLL score on the training set for different PixelSnail models on *NotImageNet32*
.

## B.2   VD-VAE

VD-VAE network is built from an encoder and decoder. In the encoder, there are regular blocks, which get an input and outputs output with the same dimension, and down-rate blocks that get input and output an output with a lower dimension. The difference between these two blocks is an avg_pool2d at the end of the down-rate block. In the decoder, there are regular blocks and mixin blocks. the regular blocks get an input and outputs output with the same dimension. The input is fed from the previous layer and the parallel layer in the encoder. The mixin block performs interpolation to a higher dimension.

Table 4: VD-VAE hyper-parameters

| Size | Encoder | Decoder |
|------|---------|---------|
| S | 32x5, 32d2, 16x4, 16d2, 8x4, 8d2, 4x4, 4d4, 1x2 | 1x2, 4m1, 4x4, 8m4, 8x3, 16m8, 16x8, 32m16, 32x20 |
| M | 32x10, 32d2, 16x5, 16d2, 8x8, 8d2, 4x6, 4d4, 1x4 | 1x2, 4m1, 4x4, 8m4, 8x8, 16m8, 16x10, 32m16, 32x30 |

In table  4 **x** means how many regular blocks are concatenated in a row. For example, 32x10 means 10 blocks in a row with a 32-channel input. **d** means a down-rate block. the number after tells the factor of the pooling. **m** means a unpool (mixin) block, for example, 32m16 means 32 is the output dimensionality with 16 layers in the mixin block.
Other hyper-parameters that were changed are EMA rate to 0.999, warm-up iterations to 1, learning rate to 0.00005, grad clip to 200, and skip threshold to 300. We used Adam optimizer with $\beta_1 = 0.9$ and $\beta_2 = 0.9$. Other hyper-parameters configure as mentioned in VD-VAE article.

## C  Supplementary models correlation measurements

In table 5 one can see that Pearson correlation is high for most of evaluation methods. This fact is consistent with the conclusion presented in the article on the ability of current evaluation methods to capture trends.

Table 5: Pearson's $\rho$ Correlation

|  | KL | RKL | FID | IS | $IS_\infty$ | KID | $FID_\infty$ | Clean FID |
|---|---|---|---|---|---|---|---|---|
| KL | 1 | 0.976 | 0.8217 | 0.7088 | 0.5656 | 0.9011 | 0.911 | 0.8962 |
| RKL | 0.976 | 1 | 0.7839 | 0.6559 | 0.5279 | 0.8552 | 0.8585 | 0.8493 |
| FID | 0.8217 | 0.7839 | 1 | 0.9441 | 0.9053 | 0.9771 | 0.9583 | 0.9829 |
| IS | 0.7088 | 0.6559 | 0.9441 | 1 | 0.9657 | 0.9047 | 0.8858 | 0.9139 |
| IS $\infty$ | 0.5656 | 0.5279 | 0.9053 | 0.9657 | 1 | 0.8301 | 0.799 | 0.8407 |
| KID | 0.9011 | 0.8552 | 0.9771 | 0.9047 | 0.8301 | 1 | 0.9825 | 0.998 |
| FID $\infty$ | 0.911 | 0.8585 | 0.9583 | 0.8858 | 0.799 | 0.9825 | 1 | 0.9863 |
| Clean FID | 0.8962 | 0.8493 | 0.9829 | 0.9139 | 0.8407 | 0.998 | 0.9863 | 1 |

In table 6 we present Spearman ranking correlation, other ranking correlation method that is similar to Kandell's $\tau$ and presented similar results.

Table 6: Spearman's $\rho$ Ranking Correlation

|  | KL | RKL | FID | IS | $IS_\infty$ | KID | $FID_\infty$ | Clean FID |
|---|---|---|---|---|---|---|---|---|
| KL | 1 | 0.9779 | 0.8449 | 0.7394 | 0.6064 | 0.9201 | 0.9353 | 0.9242 |
| RKL | 0.9779 | 1 | 0.8118 | 0.6921 | 0.5693 | 0.8828 | 0.8883 | 0.8865 |
| FID | 0.8449 | 0.8118 | 1 | 0.9238 | 0.8934 | 0.9587 | 0.9165 | 0.9627 |
| IS | 0.7394 | 0.6921 | 0.9238 | 1 | 0.9548 | 0.8904 | 0.847 | 0.8799 |
| $IS_\infty$ | 0.6064 | 0.5693 | 0.8934 | 0.9548 | 1 | 0.799 | 0.7422 | 0.7922 |
| KID | 0.9201 | 0.8828 | 0.9587 | 0.8904 | 0.799 | 1 | 0.9656 | 0.9964 |
| $FID_\infty$ | 0.9353 | 0.8883 | 0.9165 | 0.847 | 0.7422 | 0.9656 | 1 | 0.9715 |
| Clean FID | 0.9242 | 0.8865 | 0.9627 | 0.8799 | 0.7922 | 0.9964 | 0.9715 | 1 |

# D Extensive Results for Qualitative Analysis

More results regarding section 5. The datasets that are under test are CelebA and AFHQ - Dogs. In the CelebA case CLIP locates anomalies better than Inception, In AFHQ dogs dataset the performance seems more balanced.

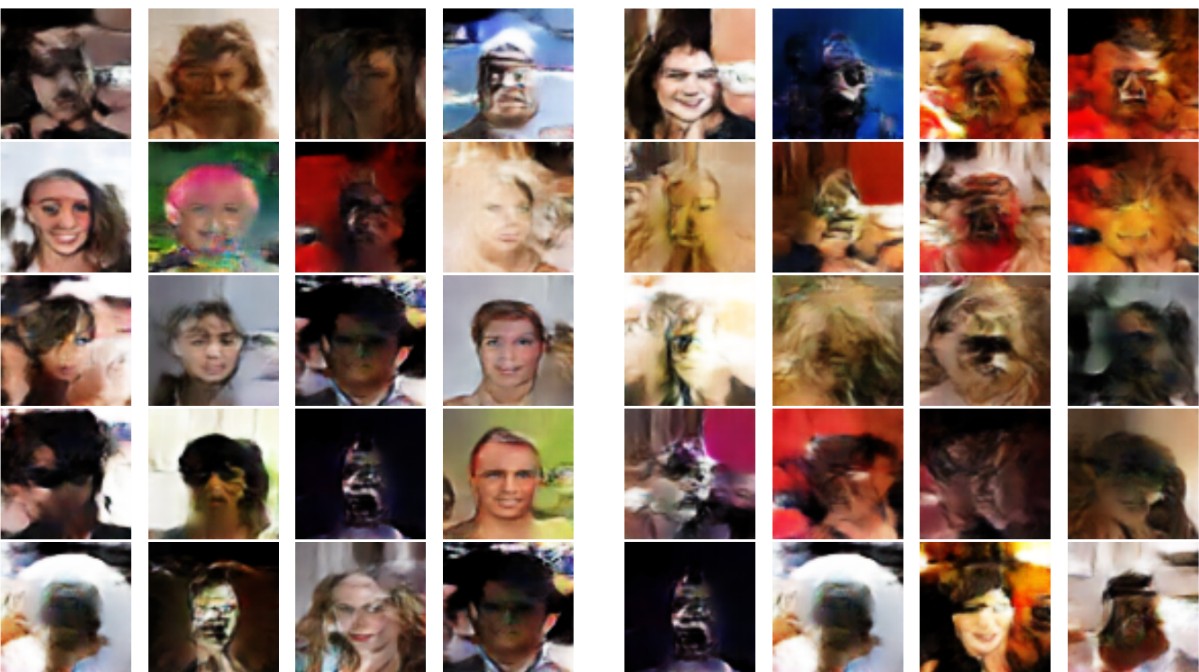

Figure 10: Inception - low probability (CelebA).  Figure 11: CLIP - low probability (CelebA).

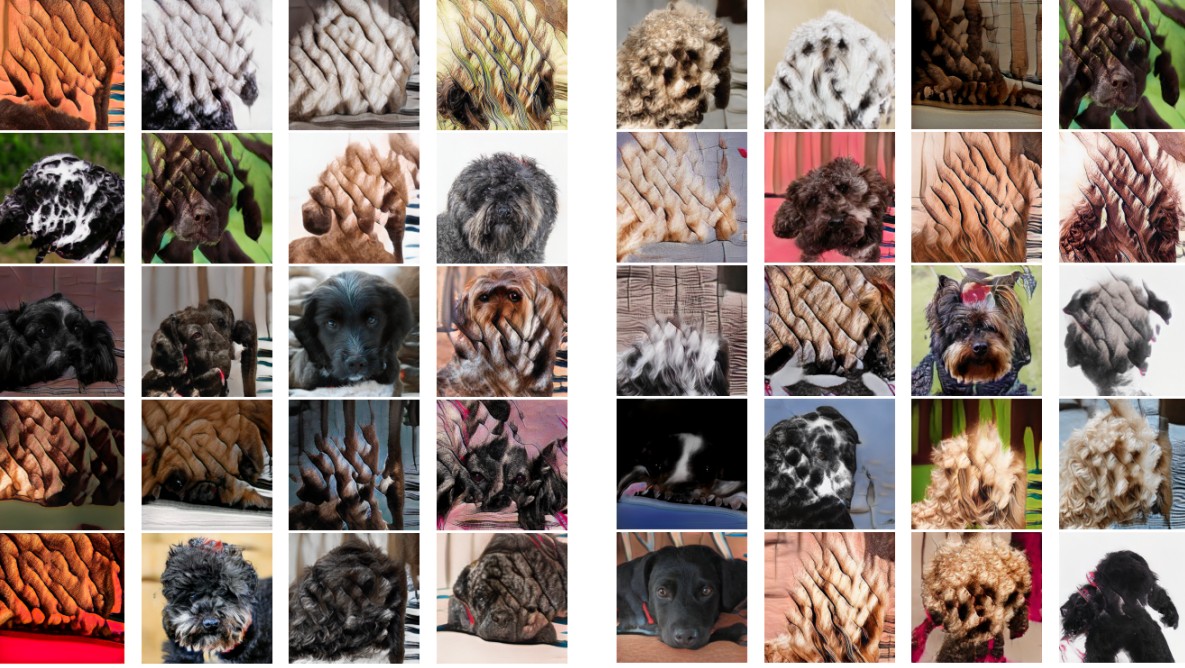

Figure 12: Inception - low probability (Dogs).  Figure 13: CLIP - low probability (Dogs).

# E    Samples Examples

These samples were generated from the different models under test in section 4, each subfigure generated on a different epoch while training the models on the ***NotImageNet32*** dataset. Figures 14 and 15 are samples from the large PixelSnail model and the medium VD-VAE model, respectively. More details on the models are in appendix B.

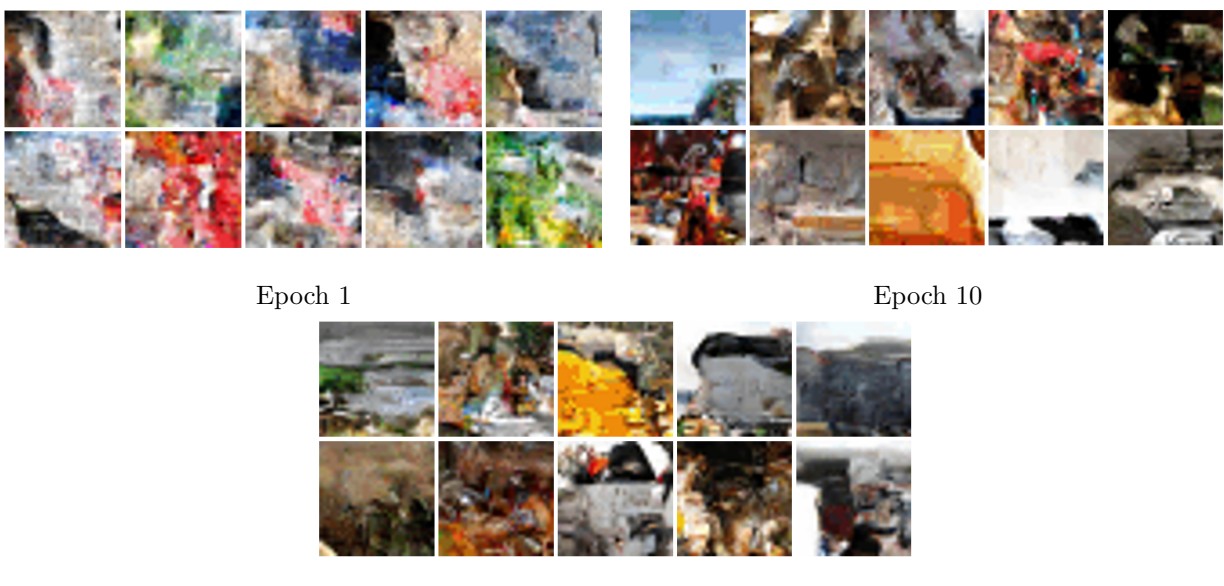

Epoch 1                                                                                    Epoch 10

Epoch 50

Figure 14: PixelSnail model samples

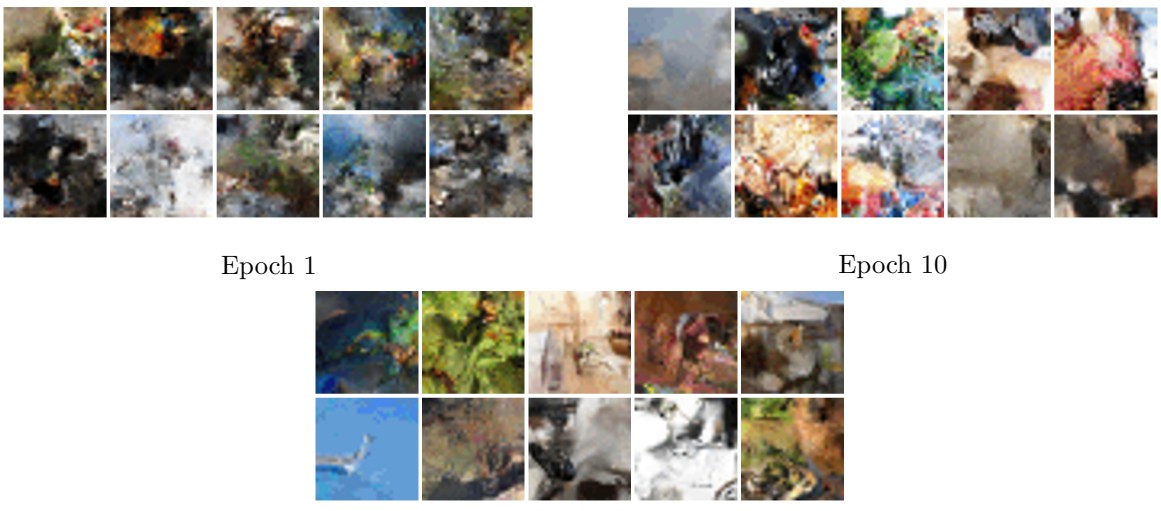

Epoch 1                                                                                    Epoch 10

Epoch 50

Figure 15: VD-VAE model samples

