# OpenReview forum: "Evaluation of Generative Models: An Empirical Study"
_TMLR — Rejected by TMLR_

### Review · Reviewer_fZYW · 2022-10-14

**Summary Of Contributions:**

The paper tackles the important problem of evaluating generative models and it is motivated by the shortcomings of existing metrics such as Inception Score (IS) and Frechet Inception Distance (FID) which are commonly used. To investigate existing metrics the authors create a dataset from which likelihoods are known (as the dataset is generated from an explicit generative model, Image-GPT). They train explicit models (PixelSnail and VD-VAE) on this dataset and compare the performance and correlation of standard divergence metrics such as KL and Reverse KL with IS and FID. They conclude by questioning the efficacy of using Imagenet features outside the imagenet domain and show that CLIP features might be better suited.

About the reviewer: I have worked on generative models research (both in implicit and explicit generative models) and have published in the area. I have a keen interest in the evaluation of generative models. I am familiar with the relevant literature.


**Audience:**

Yes

**Broader Impact Concerns:**

No concerns.

**Claims And Evidence:**

No

**Requested Changes:**

See the rooms for improvement questions in the above section.

**Strengths And Weaknesses:**

**Strengths**:

The authors present an interesting result that the KL and ReverseKL are correlated on the task they consider. Their correlation itself can be used as a metric of how well the model is able to capture the modes in the data as well as assign high likelihood to them.

The authors provide a few examples of using CLIP instead of Imagenet to compute features which compare models. The results look interesting and it can be quite applicable since CLIP is a pretrained model. The idea of using a model trained on image captioning is interesting (as supposed to standard classification models) as the feature learning signal might be stronger.

*Reproducibility*: technical details are provided in the Appendix.

**Room for improvement**

*Motivation*: The motivation of the paper can be improved. Both in the abstract and in the introductions the author start by being heavily motivated by implicit generative models. However, most of their work is not amenable to implicit generative models, which do not have likelihoods available. The authors suggest that their findings are transferable to implicit generative models, but that is not immediately clear. Explicit generative models are likely to do well on divergence tasks, since they are trained on them. The forward KL (between data and model), or a lower bound is used to train these models. That entails that when we look at Figure 3, a smooth curve there entails a smooth training landscape. The authors motivate the use of divergences (KL and reverse KL) but do not discuss their severe limitations, which have already been discussed in the literature, including in Theis et al which the authors cite (their Section 3.2 on “poor likelihood and great samples”). This has also been traditionally seen in the other direction, where models trained to minimise likelihoods obtained poor samples compared to GANs (this gap has recently been improved through novel approaches to train deep VAEs or autoregressive models, but most likely still affects the PixelSnail model referred to in this work).

The authors have to clarify:
  * Why are likelihoods the gold standard they compare against?
  * How do we know what transfers to implicit models?
  * What is the take away message beyond using other models than Inception for FID/IS/KID computation?

For using a pretrained CLIP for evaluation:
  * The authors provide qualitative results. How about quantitative results? How does a Frechet Distance compare between models when computing on Imagenet trained models vs CLIP?

*Connections to prior work*:
  * KID: when discussing KID requiring fewer samples, its worth noting that the motivation is the bias of the IS/FID. This is only mentioned later in connection to FID_\infty and IS_{\infnty}.
  * There have been existing works commenting on the use of Inception features on other datasets. In Figure 8 from “Many Paths to Equilibrium: GANs Do Not Need to Decrease a Divergence At Every Step” of Fedus et al, they use a CIFAR-10 classifier alongside the Imagenet one to compare CIFAR-10 models. The relative ranking difference is visible there as well.
  * In Are GANs Created Equal? A Large-Scale Study by Lucic et al the authors look (Figure 1) at the bias of the FID metric, its sensitivity to mode dropping as well as the sensitivity to layers and features chosen.
  * There are existing works looking at using models from data trained on generative models for evaluation (and those approaches are also suitable for implicit models). Such works include  How good is my gan? by Shmelkov et al, Training discriminative models to evaluate generative ones by Lesort et al, A classification-based study of covariate shift in gan distributions by Santurkar et al., Real-valued (medical) time series generation with recurrent conditional gans by Esteban et al,  Classification Accuracy Score for Conditional Generative Models by Ravuri et al.

*Clarity*:
Name of Section 5.1 “Quantitative analysis of the latent representation”. Do the authors mean classifier features used for evaluation? These are not latent representations and the nomenclature can be confusing in the context of generative models.

---

> ### Author Response · Authors · 2022-10-24
> **Comment for  fZYW review**
>
> Your review is greatly appreciated.
>
> Q1: Why are likelihoods the gold standard they compare against?
>
> A1: Statistical divergence measurements such as KL and Reverse KL measure well-known properties of the difference between probability density functions, as opposed to latent features of the Inception network, which what they are exactly measuring is unclear. Also, as KL and reverse KL can be considered complementary, as one is inclusive and one is exclusive, we get a better understanding by looking at both metrics. Furthermore, optimizing KL can lead to powerful models like VD-VAE, which proves it is a useful metric for generative models.
> In the end, there is no gold standard for the distance between distributions. Even in the simple small discrete case where everything is known and computable, there are many metrics to consider without any clear natural metric.
>
> Q2: How do we know what transfers to implicit models?
>
> A2: The empirical methods we examined (i.e. FID, IS, etc) are based on samples and have no access to the likelihood score of the images. As such, there is no reason to believe they will behave differently on models such as GANs
> More importantly, we show weakness in these metrics. We clearly see that they are can give very different scores to very similar models. This shows their scores can be misleading and unreliable. Even if for some reason their scores can sometimes be trusted, the fact that they can be unreliable as we have shown here raises a question about their use in the community.
>
> Q3: What is the take away message beyond using other models than Inception for FID/IS/KID computation?
>
> A3a: Empirical methods do a good job of understanding trends. Our study shows that it might be ill-suited to the fine-grained ranking of generative models, an important task for evaluation metrics.
>
> A3b: Different evaluation methods tend to disagree about the ranking order of various models. One method may consider model A better than model B, and the other will suggest the opposite. This is why researchers should use a variety of evaluation methods while examining model performances and try to achieve a high level of agreement between them.
>
> A3c: Current empirical methods are not reliable enough and we should explore alternatives. Our framework can help in this endeavor.
>
> Q4: The authors provide qualitative results. How about quantitative results? How does a Frechet Distance compare between models when computing on Imagenet trained models vs CLIP?
>
> A4: In order to address the quantitative perspective of the problem, we chose to add another section to the article in which we examined the normality assumption on the distribution of the latent features, an assumption that is on the basis of the FID method (Section [5.2]). We point out that we chose to analyze the normality test in a single dimension, rather than in a multi-dimension, for numerical stability considerations. )
>
> We thank the reviewer for his remarks regarding the connections to prior work. We added them to the text.

---

> > ### Comment · Reviewer_fZYW · 2022-10-31
> > **Thank you and detailed response**
> >
> > I thank the authors for their response and for updating the paper. Answers inline.
> >
> > > A1: Statistical divergence measurements such as KL and Reverse KL measure well-known properties of the difference between probability density functions, as opposed to latent features of the Inception network, which what they are exactly measuring is unclear. Also, as KL and reverse KL can be considered complementary, as one is inclusive and one is exclusive, we get a better understanding by looking at both metrics. Furthermore, optimizing KL can lead to powerful models like VD-VAE, which proves it is a useful metric for generative models. In the end, there is no gold standard for the distance between distributions. Even in the simple small discrete case where everything is known and computable, there are many metrics to consider without any clear natural metric.
> >
> > "as opposed to latent features of the Inception network, which what they are exactly measuring is unclear": many of the metrics, including FID do end up using a distance, but in feature space (note that as I also noted my review, "feature" space is not "latent" space in a probabilistic sense and this needs to be updated in the paper as well). While I agree that using divergences can be useful, this is taken for granted in the manuscript. Moreover, both in the manuscript and here the authors do not acknowledge the issue with likelihoods that have been highlighted by others. For example, we know that the Inception Score correlates with human evaluation and that likelihoods do not. I think this deserves much more discussion in the manuscript. Currently the manuscript does not have a clear delimitation of implicit vs explicit models, and types of evaluation available for them: while the paper starts with a strong motivation of evaluating implicit models, it does not aim to solve that issue since the KL/Reverse KL formulation is not available there. A clarification in the  text is required.
> >
> > >  A2: The empirical methods we examined (i.e. FID, IS, etc) are based on samples and have no access to the likelihood score of the images. As such, there is no reason to believe they will behave differently on models such as GANs More importantly, we show weakness in these metrics. We clearly see that they are can give very different scores to very similar models. This shows their scores can be misleading and unreliable. Even if for some reason their scores can sometimes be trusted, the fact that they can be unreliable as we have shown here raises a question about their use in the community.
> >
> > "we show weakness in these metrics.": are the weaknesses referred here the lack of correlation with likelihoods? Because if so, this is again not immediately clear to be a weakness and needs to be further clarified in the text. Otherwise, the main limitation is the usage of Inception features which has also been previously highlighted. Moreover, the KL and reverse KL are not symmetric (hence why we have two divergences and well known blind spots such as achieving high likelihood when mass is spread between modes for the forward KL) and the FID is based on the Frechet distance which is symmetric. Only from this symmetry one might achieve different correlation results. All of these aspects need to be discussed in the manuscript and clarified, in order to ensure the reader comes away with a clear picture of what is being evaluated.
> >
> > > A3c: Current empirical methods are not reliable enough and we should explore alternatives. Our framework can help in this endeavor.
> >
> > I would argue that this is only a possibility for explicit methods, where the KL (but not the reverse KL) can be approximated. This is something that is already done, and for explicit methods it is common to report both likelihoods and other metrics IS/FID. For implicit models, this is not available.
> >
> > > Q4: The authors provide qualitative results. How about quantitative results? How does a Frechet Distance compare between models when computing on Imagenet trained models vs CLIP?
> >
> > I thank the authors for doing this additional set of experiments. My understanding of the Section 5 is that tests are perform to see how Gaussian Imagenet and CLIP features are, by using a random projection into 1D space. Could the results obtained also be due to the projection methodology and the fact that Imagenet features are of a different size than CLIP features?
> >
> > For Table 2, please also add uncertainty estimates and label the model dataset column as well.
> >
> > Language: while I mentioned this also in my initial review, features are not latent codes. Please fix this throughout the manuscript, including in the new added text in section 5.2.

---

> > > ### Comment · Reviewer_fZYW · 2022-10-31
> > > **High level comment**
> > >
> > > Beyond the detailed comments I provided above, I would like to highlight the manuscript itself.
> > >
> > > For the manuscript to be clear, it needs to be restructured, and details and additional connections provided. Throughout the manuscript the questions need to be answered:
> > >   * why is this being done?
> > >   * what are the limitations of what is being done?
> > >    * what is the meaning of the results
> > >
> > >
> > > In its current state, the paper is hard to read and for a novice can be confusing, while for someone who has worked in the area it raises plenty of questions and need for additional discussion. I urge the authors to consider this and provide a significant rewrite.

---

### Review · Reviewer_Xbgg · 2022-10-27

**Summary Of Contributions:**

This paper trains several generative models on a synthetic dataset with tractable likelihoods.  Doing this enables them to measure the KL divergence (both forward and reverse) of the trained models to the true distribution they were trained on.  This enables a precise way to characterize the performance of the models and allows them to measure how well classic metrics like FID or IS compare in terms of how they would rank the models.  Overall they discover that the classic FID and IS scores don't correlate all that well with KL and suggest we as a community deprecate their use.  As alternatives they suggest switching to FID$_{\infty}$ and switching from inception features to more modern CLIP based features.  They also plan to release the dataset for use by others

**Audience:**

Yes

**Claims And Evidence:**

No

**Requested Changes:**

Wasn't really sure how to break up my review between the sections, but I made many suggestions in the previous sections.  The most important of which I've repeated here in bullet list form sorted from most to least important for securing my recommendation.

1.  Report the bpp and/or likelihood of the ImageGPT model on the original Imagnet test set.
1. Report error bars on all test statistics.
1. Revisit or remove section 5.1
1. Please include sample generations from all models trained in an appendix.
1. Report train KLs as well.
1. Fit a randomly re-initialized ImageGPT model to NotImageNet32 and compare its results

**Strengths And Weaknesses:**

Overall, I love the program presented here. By replacing the real dataset with a fake one with tractable likelihoods, we can really do true science.  I feel as though the authors if anything undersell the potential utility of such a dataset, which could be used to validate hypothesis for basically any area of machine learning.  For example, with a tractable dataset, you have access to the Bayesian optimal classification performance achievable and could evaluate classifier's by means of their KL divergence instead of just looking at their likelihood.  This gives an absolute rather than relative measure of their performance, and would really enable precise tests as to the efficiencies of various proposed modifications of networks, e.g. changing activation functions, architectures, optimizers, initialization, etc.

So, I love the program.  The paper said they plan to release the NotImageNet32 dataset, is the plan also to release the trained ImageGPT model checkpoint as well? With access to the checkpoint people would be able to generate additional samples for further analysis if desired.

That said, in any such synthetic dataset program, the primary objection will be that the synthetic dataset is not representative of real-world datasets.  There is a danger that inferences we draw on the synthetic dataset won't transfer to real-world usage.  Granted, such dangers already exist when we as a community focus on individual datasets like CIFAR-10 or Imagenet, but it is easy to be sympathetic with those who are more skeptical of a synthetic dataset being as useful as a real-world one.

Here I feel the paper could have done a better job allaying those fears. I was expecting to see an experiment where the NotImagenet32 dataset was used to train a model which would then be compared to a model trained on Imagenet train set (or a subset of the same size) and then both compared on the Imagenet test set.  I would have liked to see some measurement made of the generalization gap we open up by use of the synthetic dataset.  It seems as though in this case, the authors trained an unconditional generative model which wouldn't immediately be useful for a classifier baseline, but the generated images could always be augmented with a label distribution generated by a pretrained classifier to generate a synthetic joint $p(x,y) = p(x) p(y|x)$.  Honestly, I think it probably would have been more useful to instead of training an unconditional generative model $p(x)$ to have trained a class conditional model $p(x|y)$ which could still be used for image synthesis with a provided label distribution $p(y)$ but would make the dataset useful for a wider class of investigations. You could go wider still and train a family of class and augmentation conditioned generative models which would unlock fine-grained domain shift measurements.

In either case, what I'm trying to communicate is that a synthetic dataset is more useful the more convinced we are that it is representative of real datasets.  I feel a weakness of the paper currently is the lack of rhetorical effort put in convincing the reader that the synthetic dataset is a useful mimic of the real world. As the paper itself mentions in the introduction, another common appeal made is to human judges, I realize it's not trivial to do, but a strong rhetorical case for the utility of the synthetic dataset you generated would be if human raters weren't able to tell the difference between it and Imagenet proper.  Also, I would have liked to have seen some evaluation number for your trained ImageGPT model as well, what was its likelihood or bpp on the Imagenet test set?  Why should I believe that it is any good at modelling Imagenet?

In section 2, I'd suggest trying to add some more intuitions for what each of the metrics. For instance you could describe the Inception Score as measuring the effective number of independent classes generated, or you could point out the connection between FID and the 2-Wasserstein distance to help readers make more connections.  Please include some math describing KID in more detail. I'd also suggest giving a concrete interpretation of KL divergence.  We can interpret ~ 3/KL as the effective number of samples needed to distinguish between two models.

If I have two different hypothesis of data $H_0$ and $H_1$, after observing data I could ask how likely each of those models are to be the correct one relative to one another.  I could compute the posterior log odds of one hypothesis being correct with respect to the other as being the sum of the weight of evidence and the prior log odds.

$$ \log \frac{p(H_0|D)}{p(H_1|D)} = \log \frac{p(D|H_0)}{p(D|H_1)} + \log \frac{p(H_0)}{p(H_1)} $$

In the context here, this weight of evidence is the log density ratios of the two models, model and data ($H_0$ = "model is true", $p(D|H_0) \to p_{\text{model}}(D)$).  If I started with prior log-odds of 0 (even odds) if I accumulated ~ 3 nats of evidence I would have roughly 20 to 1 odds in favor of one of the hypotheses, and under usual statistical conventions I'd say I could distinguish which was correct.  The KL divergence is the expected weight of evidence under one of the models.  So $KL[p;q]$ is the expected weight of evidence if we actually drew samples from $p$, and we could interpret $3/KL[p;q]$ as the effective number of samples we would have to observed from $p$ on average before we could convince ourselves that they really did come from $p$ rather than $q$.  Similarly, $3/KL[q;p]$ we could interpret as the effective number of samples from $q$ on average we'd have to observe to convince ourselves that $q$ was a better model that $p$ for those samples.

With this understanding, one of the things that gives me the most pause is just how poor the models you fit do.  In figure 3 you are reporting $KL[p_\text{data}; p_{\text{model}}]$ values of nearly 200 nats. This is a very large gap.  Its suggesting that (ignoring issues of independence) that after observing only roughly 1% of one generated image we should be able to distinguish between the model and the data.  Figure 2 shows samples from your image-GPT, but I would have liked to have seen samples generated by the trained PixelSnail and VDVAE models.


Another thing I would have liked to have seen are KL divergences measured on the training set.  You could compute the expected log density ratio but on the training set images:
$$ \sum_i \log \frac{p_{\text{data}(x_i)}}{p_{\text{model}(x_i)}} $$
This would tell us how well each of the models did at optimizing for their training objective.  I don't know whether the 200 nat KL is being driven by the a gap introduced by having only been trained on 30k images, or if its a modelling gap.  Seeing the training KL would let me assess whether these models match the training distribution on the empirical distribution or whether they have failed to model the training set.

Why didn't you train a randomly re-initialized ImageGPT model on the NotImageNet32 dataset?  Wouldn't this be an interesting comparison as it would be from the same model family as the generative process? This would let us know how large of a generalization gap we should expect from the other models due to the finite samples in the training set.  While we're at it, in a perfect world I would have liked to have seen an independently trained ImageGPT model, trained on Imagenet itself also evaluated for its performance on the NotImageNet32 test set.

70k training images feels a bit small for a training set.

In this setup, evaluating things on the test set is an unbiased monte carlo estimate of the expectations with respect to the true (synthetic) data distribution, but you can also use the observed variance of the statistics on the test set to assess standard errors.  I don't know if the instabilities in FIG and IS over the training trajectory are real or whether they are just fluctuations in the sampling noise of the 30k test set. Please include error bars on Figures 3, 4, and 5.

There is a flaw with Section 5.1.  Probability density isn't the same as mass.  For a high dimensional standard Gaussian, the point with the highest likelihood is the origin, but this sample is exponentially unlikely to ever be observed when sampling from the distribution.  Instead samples from a high dimensional Gaussian distribution have norms that are concentrated at $\sqrt D$.  For this reason its not a good idea to use likelihood for OOD detection.  See for example [Nalisnick et al. 2019](https://arxiv.org/abs/1810.09136), [Hendrycks et al. 2019](https://arxiv.org/abs/1812.04606) or [Choi et al. 2019](https://arxiv.org/abs/1810.01392) or a useful explainer in [Caterini and Loaiza-Ganem 2021](https://arxiv.org/abs/2109.10794).  Also, why are Figures 6 and 7 showing samples from a pre-trained StyleGAN2-ADA and not any of the generative models trained for this work?  Or, relatedly, why didn't you evaluate the KL, FID and IS of the StyleGAN2-ADA model on the NotImageNet32 test-set?

For section 5.2, I don't think we should be slaves to classical statistics. In addition to the reported p-values, which don't carry a lot of rhetorical weight with me because I'm not familiar with the details of the K-squared normality test, why not also just show some histograms of the resulting one dimensional projections.  With p values so ridiculously low, I suspect it would be very easy for me to *see* that random 1 dimensional projections of the features were non-gaussian and this would carry much more rhetorical weight.  Or consider something like a [Quantile-Quantile Plot](https://en.wikipedia.org/wiki/Q%E2%80%93Q_plot) to visually demonstrate the disagreement.


Also, I feel as though you sell the utility of this approach a little short with regards to evaluating implicit generative models.  You could always evaluate the likelihood of a GAN's generated images under your generative model (provided you properly take care to interpret the numbers in light of the discussion above regarding likelihoods and masses), or you could try more exotic tricks from the literature to either train GANs that admit likelihood calculations, e.g. [Dieng et al. 2019](https://arxiv.org/abs/1910.04302) or use variational bounds or other tricks to estimate some likelihood based quantities, e.g. [Alemi and Fischer 2018](https://arxiv.org/abs/1802.04874)

 * misspelling, test-bad -> testbed just before Section 4 starts.
 * typo in Table 1 caption, Kendall not Kandell

Overall, I really like the program presented here. I love the idea of using synthetic datasets to evaluate the performance of our models, or more generally validating the hypotheses we have about machine learning.  This paper is a great first step for that kind of program.  As for weaknesses, I feel as though it falls a bit short, especially with regards to convincing readers that the synthetic dataset here is of interest in that we could expect inferences drawn on this dataset to carry over to the real world.  I also feel as though several of the evaluations done are lacking, such as missing errors on the computed test-set metrics and the sort of flaw with section 5.1, using likelihood as a measure of OOD at all

---

> ### Author Response · Authors · 2022-11-05
> **Comment for Xbgg Review**
>
> Your review is greatly appreciated.
>
> Q1: I feel as though the authors if anything undersell the potential utility of such a dataset, which could be used to validate hypothesis for
> basically any area of machine learning.
>
> A1: This is a statement we agree with. We felt, however, that we had to be cautious and only declare facts that we had verified.
>
> Q2:  For example, with a tractable dataset, you have access to the Bayesian optimal classification performance achievable and could evaluate classifier's by means of their KL divergence instead of just looking at their likelihood.
>
> A2: Future research in this area is worthwhile, even though simple classification problems with a single valid label have a zero optimal Bayes loss.
>
> Q3:  Is the plan also to release the trained ImageGPT model checkpoint as well?
>
> A3:  We used the publicly available ImageGPT model weights.
>
> Q4:  The primary objection will be that the synthetic dataset is not representative of real-world datasets.I feel the paper could have done a better job of allaying those fears.
>
> A4: We will add to the article more information regarding ImageGPT capabilities. ImageGPT is a likelihood model and it is a strong model. Also, we supply samples of NotImageNet32 in the article to address this subject.
>
> Q5:  It seems as though in this case, the authors trained an unconditional generative model which wouldn't immediately be useful for a classifier baseline.
>
> A5:  As DNNs are sensitive to imperceptible changes, the classification does not guarantee perceptual quality. Adversarial attacks on input are an example of such behavior.
>
> Q6:  I would have liked to have seen some evaluation number for your trained ImageGPT model as well, what was its likelihood or bpp on the Imagenet test set? Why should I believe that it is any good at modelling Imagenet?
>
> A6: We will add information about the likelihood score to the article. Could you please explain what bpp score means? This term is unfamiliar to us.
>
> Q7:  I would have liked to have seen samples generated by the trained PixelSnail and VDVAE models.
>
> A7:  We will Add photos from pixelsnail and VDVAE to the appendix, along with an intuitive explanation about different metrics.
>
> Q8: Why didn't you train a randomly re-initialized ImageGPT model on the NotImageNet32 dataset?
>
> A8: We deliberately used weaker models than ImageGPT, to make our models weaker then the actual distribution they are fitting. This should be similar to the real case.
>
> Q9: 70k training images feel a bit small for a training set.
>
> A9: NotImageNet32 dataset size is similar to CIFAR10, which is widely popular. we haven’t created an order of magnitude larger dataset due to computation limitations, and due to the long sampling time of autoregression models.
>
> Q10: Please include error bars on Figures 3, 4, and 5.
>
> A10:  To assess the variance of the results we used the Jack-Knife sampling method. The error bar was small (1E-7 scale) and unnoticeable.  We will mention this fact in the article and include some variance of the results in the appendix.
>
> Q11:  For a high dimensional standard Gaussian, the point with the highest likelihood is the origin, but this sample is exponentially unlikely to ever be observed when sampling from the distribution.
>
> A11: While the mode is unlikely to be sampled, that is true to any specific point. Both approaches consider who are far from the mean (compared to the std) as outliers. We did not claim these are the only outliers, but if the model was Gaussian then these would be.
>
> Q12: For this reason its not a good idea to use likelihood for OOD detection.  relatedly, why didn't you evaluate the KL, FID, and IS of the StyleGAN2-ADA model on the NotImageNet32 test-set?
>
> A12:  In IS/FID the Inception backbone, as well as NotImageNet32, trained on ImageNet. In this part, we wanted to assess the ability of a CLIP Vs. Inception backbone networks to model the distribution of another, non ImageNet, based dataset, due to the fact that FID and IS are usually used as metrics on non-ImageNet datasets. We did include, however, an assessment of NotImageNet32 in section 5.2.
>
> Q13:  [Section 5.2] I suspect it would be very easy for me to see that random 1 dimensional projections of the features were non-gaussian and this would carry much more rhetorical weight.
>
> A13: These are effective as a normality test, and prove our claim in a quantitative manner. We accept your offer and will include histogram plots in the article.
>
> Q14: I feel as though you sell the utility of this approach a little short with regards to evaluating implicit generative models.
>
> A14:  We agree with your intuation but wanted to remain cautious about the assertions we make about things we haven’t verified. About your suggestion regarding the method to evaluate GANs likelihood, the problem lay in the fact that you can’t be sure if your model is problematic or your likelihood assessment, so we focus on models that we can be positive regarding their results.

---

> > ### Comment · Reviewer_Xbgg · 2022-11-10
> > **Response to Comment**
> >
> > Q2: I think you'll discover that if you train a class conditional generative model, it will not be degenerate in the sense of assigning 100% probability to a single class. I also strongly feel as though you shouldn't think of real world datasets as having this property.
> >
> > Q3: You should make that much more explicit in the paper and point people to the checkpoints.
> >
> > Q6: By bpp I meant 'bits per pixel', a common set of units for reporting likelihood for generative models. It's often reported as bits per pixel depth in the sense that an uncompressed image would have 8 bits per pixel depth, while good generative models on CIFAR10 have < 3 bpp.  See e.g. https://paperswithcode.com/sota/image-generation-on-cifar-10
> >
> > Q10: Honestly, I'm very skeptical.  You're saying that across your test set you have ~10^{-7} variation in your computed KLs?  That doesn't make any sense to me. I've never seen a generative model with such consistent KLs, I'm very strongly suspicious there's been a mistake.
> >
> > Q11: You are correct that the probability of sampling any individual point is always small.  It is also true that likelihood does not make for a good measure of out of distribution, for the reasons I outlined and contained in the references I gave.  High likelihood does not mean that a sample is representative of typical draws from the distribution.  The highest likelihood points of a high dimensional gaussian are all points near the origin with small norm, meanwhile nearly all of the random draws you observe will have norms tightly clustered around $\sqrt D$.  If all I did was look at likelihood, I would believe that samples from a 100 dimensional distribution with mean 0 and diagonal deviations of $10^{-3}$ were all in distribution for a 100 dimensional distribution with mean 0 and standard deviation 1, while the KL from the first to the second is very large, ~640 nats.
> >
> > It sounds like you've agreed to incorporate a number of the suggestions I made.
> >
> > I look forward to seeing the revised manuscript.

---

### Review · Reviewer_LFVp · 2022-11-05

**Summary Of Contributions:**

The work offers insight into popular generative model evaluation metrics, including the inception score and Fréchet Inception Distance. In particular, they show that the correlation between these metrics and (reverse) KL-divergence is not always high, and that using CLIP embeddings over Inception embeddings could provide more reliable evaluation.

**Audience:**

Yes

**Broader Impact Concerns:**

No concerns

**Claims And Evidence:**

Yes

**Requested Changes:**

Please see the weaknesses above. In particular, I would ask the authors to address the major weaknesses concerning the use of (R)KL and the reliability of results (error bars and qualitative study).

**Strengths And Weaknesses:**

**Strengths**:

1. Addresses generative model evaluation, an important and unsolved problem. Relevant for community and some novel and useful recommendations (in Conclusion).

2. Overall well-written and presented

3. The design of the evaluation procedure (training two consecutive explicit likelihood-based generative models) is clearly useful for *metric evaluation* itself, for having both a ground-truth density and relatively realistic data. The latter provides benefits over entirely synthetic toy datasets that are often seen in literature. That being said, the access to the ground-truth density is only used for computing (R)KL, which can be considered quite limiting (see Weakness 1).

**Weaknesses**:

1. Method, specifically choice for KL and RKL as ground truth quality. Why the choice for (R)KL divergence as ground truth metric to compare against? As far as I’m aware, (R)KL is not good at measuring diversity. As a result, I wonder whether the conclusions drawn from this analysis really indicate any metric is better or more reliable---e.g. (p. 8) *“Among the Inception-based metrics, FID∞ has the highest correlation with KL and RKL which indicates that it is a more *reliable* metric than the other.”*
    This question also relates to (p.6) *“KL-Divergence has been thoroughly investigated in the fields of probability and information theory, and their properties along with what they measure are well known. Thus, comparing them to heuristic methods such as FID will shed light on these empirical methods”*, but despite this last claim it is unclear what properties of (R)KL are used for "shedding light" on the evaluated metrics.

2. Errors/standard deviations. For understanding reliability of results and metrics themselves better, it would really help to use multiple runs and add standard deviations, especially in Figures 3 and 4.

3. Qualitative analysis reliability. Though CLIP indeed seems more suitable for the Wild class than Inception, this is not observed for the dogs. Is CLIP generally better, as you conclude? You mention (p.9) *“While this is only a small number of images, we see similar behaviour across various non-ImageNet datasets”*, but these results are not included.

4. Section 5.2 motivation. It is indeed clear from this section that CLIP follows a Gaussian better than Inception. However, it is poorly motivated why non-gaussianity inherently implies it is a less useful metric for generative modelling. It would have been useful to have a quantitative comparison of FID vs F(CLIP)D in terms of downstream quality.

5. Related work could be better. Though the authors cite (Sajjadi, 2018) and (Kynkäänniemi, 2019), it is unclear why these metrics are not used for evaluation---especially since these could help with Weakness 1, measuring more than just “how much does it correlate with (R)KL”.

*Minor*:

1. Typo (p. 6) test-bad

2. Exclusivity and inclusivity. You mention KL and RKL are good for “exclusive” and “inclusive” evaluation, but it would be useful to have some references here.

3. $FID_{\infty}$. Though Section 2 covers all metrics that are included in the experiments, readability would be aided by briefly explaining how $FID_{\infty}$ and $IS_{\infty}$ are less-biased---especially since one of the recommendations is to use $FID_{\infty}$ more.

5. Captions could be significantly improved---right now some Tables and figures are not self-explanatory. This includes:

    1. Table 1 (+place above table);

    2. Table 2;

    3. Figures 6-9 (perhaps merge into one figure and use one main caption to elaborate on experiment)

    4. Appendix C tables

---

### Decision · Action_Editors · 2023-01-25

**Recommendation:** Reject

**Comment:**

The paper studies the problem of evaluating generative models.

It investigates shortcomings of existing metrics (IS, FID) by creating a dataset where likelihoods are known and measuring divergence metrics such as KL and Reverse KL.

Overall, the reviewers felt that the current version falls short of the acceptance threshold and the consensus decision leaned towards rejection for the following reasons:
- Some of the claims are not supported by the empirical evidence: see concerns about limitations of likelihood estimates from Reviewers Xbgg and fZYW, and Reviewer LFVp's comments about claims about FID vs IS.
- Writing and motivation could be improved: e.g. the paper spends a fair bit of time talking about implicit models, however likelihoods are not available for most implicit models, so it's not possible to evaluate divergences directly
- Add more details on experimental setup and reproducibility as some of the error bars seem suspiciously low.
- Discussion of related work needs to be improved (e.g. why are some of the relevant work not considered for evaluation here?)

I recommend rejection.

**Audience:**

Yes. The paper is relevant to TMLR.

**Claims And Evidence:**

No. Some of the claims are not well supported by evidence (see comment below)